# Unraveling *NPR*-like Family Genes in *Fragaria* spp. Facilitated to Identify Putative NPR1 and NPR3/4 Orthologues Participating in Strawberry-*Colletotrichum fructicola* Interaction

**DOI:** 10.3390/plants11121589

**Published:** 2022-06-16

**Authors:** Yun Bai, Ziyi Li, Jiajun Zhu, Siyu Chen, Chao Dong, Qinghua Gao, Ke Duan

**Affiliations:** 1College of Food Science, Shanghai Ocean University, Shanghai 201306, China; baiyun18521326378@163.com (Y.B.); csy13023156961@163.com (S.C.); 2Shanghai Key Laboratory of Protected Horticultural Technology, Forestry and Fruit Tree Research Institute, Shanghai Academy of Agricultural Sciences (SAAS), Shanghai 201403, China; 2016204013@njau.edu.cn; 3Shanghai Institute of Technology, Ecological Technique and Engineering College, Shanghai 201418, China; lzy18185768772@163.com (Z.L.); zhujiajun96@163.com (J.Z.)

**Keywords:** strawberry, *NPR*-like family, phylogeny, expression profile, anthracnose

## Abstract

The salicylic acid receptor NPR1 (nonexpressor of pathogenesis-related genes) and its paralogues NPR3 and NPR4 are master regulators of plant immunity. Commercial strawberry (*Fragaria* × *ananassa*) is a highly valued crop vulnerable to various pathogens. Historic confusions regarding the identity of *NPR*-like genes have hindered research in strawberry resistance. In this study, the comprehensive identification and phylogenic analysis unraveled this family, harboring 6, 6, 5, and 23 members in *F. vesca*, *F. viridis*, *F. iinumae*, and *F.* × *ananassa*, respectively. These genes were clustered into three clades, with each diploid member matching three to five homoalleles in *F.* × *ananassa*. Despite the high conservation in terms of gene structure, protein module, and functional residues/motifs/domains, substantial divergence was observed, hinting strawberry NPR proteins probably function in ways somewhat different from *Arabidopsis*. RT-PCR and RNAseq analysis evidenced the transcriptional responses of *FveNPR1* and *FxaNPR1a* to *Colletotrichum fructicola*. Extended expression analysis for strawberry *NPR*-likes helped to us understand how strawberry orchestrate the NPRs-centered defense system against *C. fructicola*. The cThe current work supports that FveNPR1 and FxaNPR1a, as well as FveNPR31 and FxaNPR31a-c, were putative functional orthologues of AtNPR1 and AtNPR3/4, respectively. These findings set a solid basis for the molecular dissection of biological functions of strawberry *NPR*-like genes for improving disease resistance.

## 1. Introduction

The sessile plants are constantly threatened with various biotic and abiotic stresses. They have evolved multiple defense mechanisms to ensure survival and development. The expression of pathogenesis-related (*PR*) genes in distal uninfected tissues is commonly a part of systemic acquired resistance (SAR) after infection with pathogens. The *npr1* (NONEXPRESSOR OF PATHOGENESIS-RELATED GENES 1) mutant was first identified in *Arabidopsis* with reduced resistance [1]. NPR1 was characterized as a protein consisting of a N-terminal BTB domain, followed by ankyrin repeats and C-terminal nuclear localization signal [2]. Several cysteine residues within and beyond the BTB domain contribute to the reversible structural switching of AtNPR1 upon redox changes, and monomeric NPR1 is translocated to the nucleus to regulate gene expression [3]. NPR1 was confirmed to be a positive regulator of plant defense, while its paralogues NPR3/4 were largely the negative regulators of plant defense [4]. SUMOylation of NPR1 promotes its phosphorylation and interaction with TGAs to facilitate PRs expression [5]. Both NPR1 and NPR4 are required for the activation of N-hydroxypipecolic acid biosynthesis to induce SAR, pattern-triggered immunity (PTI), and effector-triggered immunity (ETI) [6]. NPR protein turnovers and their binding with salicylic acid (SA) are finely tuned by the endogenous SA level, which is crucial for the transcriptional reprogramming underlying plant immunity [7]. The cytosolic AtNPR1 oligomers could function as a molecular chaperone to protect plants from stress [8]. Recently, the structural basis of SA binding by NPR proteins has been unveiled [9]. These extraordinary pioneer works have reinforced the central regulatory roles of NPR1 and NPR3/4 proteins in plant defense. 

The NPR-like family is classified into three clades meeting with phylogenetic analysis and functional diversification [10]. As the important transcriptional cofactors, NPR1 and its homologues have been identified in many plants [9,11,12,13,14,15,16,17,18,19,20]. Beyond plant immunity, the roles of the NPR family have extended to the development of lateral organs such as leaves and flowers, the regulation of the circadian rhythm, and proteins in the endoplasmic reticulum [21].

Strawberry is cultivated worldwide for its attractive fruits. The commercial strawberry (*Fragaria* × *ananassa*) is an allo-octoploid perennial herb crop of the *Rosaceae* family, one of the youngest domesticates originating in early eighteenth-century Europe [22]. Four diploid species, *F. vesca*, *F. iinumae*, *F. nipponica*, and *F. viridis*, have been proposed as the progenitors of *F.* × *ananassa*, and *F. vesca* has been accepted as the dominant subgenome progenitor [23]. Improved genomes and annotations of *F. vesca* and *F.* × *ananassa* have opened the door for breakthroughs in strawberry gene functional study in the near future [24,25]. The reference genomes of five more strawberry diploid species, including *F. viridis*, have been available [26]. Although controversy about the diploid progenitors to C and D subgenomes remains [27,28], it has been recently proposed that the ancestors of subgenomes C and D in *F.* × *ananassa* might be extinct, and *F. nipponica* and *F. viridis* could still serve as the closest living C and D relatives [29]. This landmark progress enables us to distinguish between strawberry genes and their homoalleles at the chromosome scale for the first time.

Strawberry is vulnerable to various pathogens, which have endangered both ecological and food security. Since growing resistant crops is the most economical and eco-friendly strategy for disease management, genetic engineering with *NPR* genes to increase crop resistance has been utilized in many crops [30,31,32]. The overexpression of the positive regulator of plant immunity AtNPR1 conferred broad-spectrum disease resistance but negatively influenced plant growth in strawberry [33]. However, the identity of strawberry NPR1 protein is still a mystery, and the phylogeny of strawberry *NPR*-like family genes in different species remains unsolved. A hypothetical model including FaNPR1 was proposed for the partially activated SA-dependent defense in strawberry against anthracnose causal agent *Colletotrichum acutatum* [34]. In our previous work, the octoploid strawberry homoallele of *NPR*-like *F. vesca* gene20070 was named as *FaNPR1*, which showed an early expression induction with fluctuations in cv. ‘Jiuxiang’ infected with *C. fructicola* [35,36]. The same gene20070 was named as *FvNPRL-1*, whose ectopic expression in *Arabidopsis* resulted in suppressed resistance to *Pseudomonas syringae* pv. Tomato DC3000, hinting at the potential functional ortholog of *Arabidopsis* NPR3/4 [14]. The homoalleles of *FveNPR-like* (gene28768) were named after *FaNPR1* and qPCR analysis revealed an consistent expression induction of *FaNPR1* in strawberry cv. ‘Benihoppe’ after exogenous SA treatment or *Podosphaera aphanis* infection, the causal agent of powdery mildew [37]. Later, three homoalleles (FxaC_10g12880, 11g10120, and 12g38540) of *FvNPRL-1* (gene20070) were named as *FaNPR3-likes*, whose transcription responses to *P. aphanis* were again confirmed via RNA-seq data [38]. The identities of putative NPR1 and NPR3/4 functional orthologues in strawberry remain confusing. 

Identifying genes important for plant immunity and exploring the defense mechanisms may provide valuable resources for resistance breeding in strawberry. Concerning the importance of NPR-like proteins and previous confusions over strawberry *NPR*-like genes, there is an urgent requirement to identify this gene family in diploid and octoploid strawberry. The specific goals of the current work are to (i) systematically identify and distinguish the *NPR*-like family in cultivated octoploid strawberry (*F.* × *ananassa*) and its relatives (*F. vesca*, *F. iinumae*, *F. nipponica*, *F. viridis*, and *F. nilgerrensis*), (ii) characterize the expression responses of whole *NPR*-like family to *C. fructicola* infection in *F. vesca* and *F.* × *ananassa*, and (iii) initially propose the NPR-centered signaling network underlying strawberry against *C. fructicola* based on in silico protein–protein interacting (PPI) prediction and RNA-sequencing data. For the first time, we confirmed the expression of strawberry *NPR*-like genes of clade I (AtNPR1/2) and clade III (AtNPR5/6). The molecular features and expression patterns revealed in the current work integrated with previous reports enabled us to recognize the potential orthologues of NPR1 and NPR3/4 in strawberry. The current work provides a solid basis for further functional study of *NPR*-like genes in commercial strawberry and its relatives.

## 2. Results

### 2.1. Diversity and Phylogeny of NPR-like Members in Fragaria Species

Based on BLAST search with AtNPRs as queries and monitoring the presence of conserved domains via PFAM/CDD analysis, a total of 48 putative *NPR*-like genes were identified from five diploid and one octoploid *Fragaria* genomes. Specifically, there were 23 loci in octoploid *F.* × *ananassa*, six in diploid *F. vesca*, six in *F. viridis*, five in *F. iinumae*, three in *F. nipponica*, and five in *F. nilgerrensis* (Table 1). It was worth noting that the information of *NPR*-like genes in *F. nipponica* was incomplete due to the absence of high-quality genome annotation. The isoelectric points (pI) of strawberry NPR-like members were less than 7.0, ranging from 5.29 (FxaNPR32f) to 6.94 (FxaNPR32d). The molecular weights varied between 12.1 (FxaNPR32f) and 66.3 kDa (FnilNPR32, FvirNPR32a, and FxaNPR32d). PSORT prediction suggested the subcellular localization of 16 strawberry NPR-like proteins in chloroplast and 12 in endoplasmic reticulum. Strikingly, there were 25 *Fragaria* NPR-like proteins predicted in cytoplasm, 25 proteins in nucleus, and 13 members potentially as shuttles between nucleus and cytoplasm or plasma membrane, which hints at the versatile nature of NPRs, especially in signaling and expression regulation. 

To understand the evolutionary relationships among NPR-likes from diverse *Fragaria* species and *Arabidopsis*, an unrooted neighbor joining phylogenetic tree was constructed using the protein sequences of 48 strawberry NPR-like members and six AtNPRs (Figure 1). This analysis supported the division of all proteins into three clades (Clade I, Clade II, and Clade III), with each clade headed by two *Arabidopsis* orthologs. Accordingly, strawberry NPR-like proteins were named following the nomenclature of *Arabidopsis* orthologs in certain clade. Each NPR-like protein in diploid *Fragaria* species corresponds to three to five homologous proteins in octoploid *F.* × *ananassa*. Clade I NPR-like proteins were not identified in *F. nipponica*. Moreover, the percentage of identical amino acids shared among *Fragaria* NPR-like proteins and AtNPRs was listed in Appendix A (Appendix A for all 54 members, Appendix A for Clade I–III). Among all 54 members, the percentage of identity in amino acids ranged from 12.5% to 100%. Among clade I, II, and III, the common identity varied between 44.78 and 99.32%, 25.81 and 99.66%, 31.02 and 100%, respectively. The percentage lower than 40% largely occurred with the truncated members predicted in *F. nipponica*. 

Interestingly, all *Fragaria* NPR-like members of clade I were predicted as shuttles between the nucleus and cytoplasm, and all members of clade III beyond FnipNPR5 might share locations in chloroplast and endoplasmic reticulum. Members of clade II were predicted in many subcellular locations, including the cytoplasm, endoplasmic reticulum, chloroplast, and nucleus. Three members of clade II might also be shuttles between the nucleus and cytoplasm (FveNPR31) or plasma (FxaNPR32e and FvirNPR32b). Furthermore, pI of clade I members varied between 6.17 and 6.74, while pI of all clade III members except for FnipNPR5 ranged from 6.20 to 6.30. Clade II could be grouped into three subclades. pI of NPR31-likes is between 6.15 and 6.69, and of NPR33-likes between 5.55 and 5.93. NPR32-like members from all diploid species beyond *F. vesca* have a pI ranging from 5.47 to 5.88. *F.* × *ananassa* and its dominant sub-genome progenitor *F. vesca* possess NPR32-like members with pI ranging from 6.07 to 6.94, with the exception of FxaNPR32f (5.29).

### 2.2. Structural Features of Fragaria NPR-like Members in DNA and Amino Acid Sequences

The phylogeny trees among *Fragaria* NPR-like members generated based on amino acid sequences (Figure 1) and those based on nucleotide sequences in the coding region (Figure 2) were largely identical with the exception of FnilNPR5. Generally, the exon–intron organization of strawberry *NPR*-like genes were conserved within a certain clade. All genes of clade I beyond *FvirNPR1* and *FxaNPR1e* contain four exons separated by three short introns, with the second and third exons being the longest and shortest, respectively. Surprisingly, both *FvirNPR1* and *FxaNPR1e* possess five exons with a short exon (second) followed by an extraordinarily long intron. *NPR*-like genes in clade III uniformly possess two exons, and the relative length of intron and exon in coding region is conserved. For clade II, most genes contain four exons of relative length similarly observed in clade I, although variations in intron sizes occur. The subgroup represented by *NPR31*-like genes possesses three introns of a similar length. The first intron in *NPR32a*-like genes is longer than that in *NPR31*-likes. Both the first and the third introns in *NPR33*-likes genes are longer than those in *NPR31*-likes. Five *Fragaria NPR32b*-like genes, including *FveNPR32b*, *FvirNPR32b*, and *FxaNPR32e-f-g*, similarly consist of three exons, which was not observed in *Arabidopsis*. In addition, the predicted *FnipNPR5* and *FnipNPR33* consist of one and two exons, respectively, which again triggered doubt about the accuracy of annotation via FGENESH in Softberry for *F. nipponica* genomic DNA. 

An analysis of protein domain module revealed that most *Fragaria* NPR-like proteins contain a typical N-terminal Bric-a-brac, tramtrack, broad-complex (BTB)/poxvirus, zinc finger (POZ) domain (BTB/POZ), Ankyrin repeats (Ank), and a NPR1/NIM1 like defense protein C terminal (NPR1_like_C) (Figure 3). The Ankyrin repeats domain is absent in FxaNPR1e of clade I. With the exception of FnipNPR5, all proteins of Clade III have no NPR1_like_C domain. The aforementioned five *NPR32b*-like genes composed of three exons encode proteins only possessing NPR1_like_C domain. Of the three NPR-like members deduced in *F. nipponica*, FnipNPR31 is the unique member possessing all three functional domains, while FnipNPR33 and FnipNPR5 only contain Ankyrin repeats and NPR1_like_C domain without BTB/POZ domain. The potentially misleading grouping of FnipNPR5 in phylogenetic clade III might result from the presence of a complete ANK domain along with a partial NPR1_like_C domain in this candidate.

Comparing the motif compositions in NPR1 and NPR-like proteins between *Fragaria* spp. and *Arabidopsis* revealed the presence of *Fragaria*-specific motifs at the N-terminal before BTB/POZ domain in most members of clade I and clade II (Appendix A). Multiple sequence alignment showed that *Fragaria* NPR-like proteins of clade I possess a variant of IkB-like phosphodegron motif (DSxxxxS) (Appendix A) different from the motif ‘DSxxxS’ in *Arabidopsis*, which was related to the degradation of NPR1 [4]. Six cysteine residues (C82, C150, C155, C160, and C216) beyond C156 crucial for NPR1 oligomer formation [3] were conserved in five NPR1-like members identified in octoploid strawberry and each one in four diploid species. C156 in *Arabidopsis* NPR1 was substituted by a serine residue in all *Fragaria* NPR1-likes beyond that in *F. nilgerrensis* (alanine residue). Virtually, clade I NPR-like proteins in *Fragaria* contain nearly identical Ankyrin repeats, the NIMIN1/2-binding region, the nuclear localization sequence NLS, and the SUMO-interaction motif (SIM) [2,5]. The putative hinge region (LENRV) for SA binding is conserved in all clade I NPR-likes except for FiinNPR1 [39]. In addition, this region is also conserved in all members in *Fragaria* NPR31 and NPR33 subgroups but not in NPR32 subgroup.

### 2.3. Physical Distribution of NPR1-like Loci in the Genomes of F. vesca and F. × ananassa

The closest diploid relatives of octoploid strawberry have been suggested as *F. vesca*, *F. iinumae*, *F. nipponica*, and *F. viridis* for A, B, C, and D sub-genomes, respectively [29]. There is no doubt that *F. vesca* is the dominant diploid ancestor of *F.* × *ananassa* [23]. However, the sub-genome assignment problem in *F.* × *ananassa* has persisted [26,28,29]. The current comparison of the chromosomal location of *NPR*-like genes was limited to *F. vesca* and *F.* × *ananassa*. 

*NPR*-like family genes are unevenly dispersed on three chromosomes of *F. vesca* (Figure 4A). *FveNPR1* of clade I is on Fvb1, and its homoalleles in octoploid strawberry are composed of five loci located in all four sub-genomes, with one more allele in B sub-genome (Figure 4B). Two thirds *NPR*-likes genes are located on Fvb6 of the *F. vesca* genome. The *FveNPR5* encoding protein of clade III has four homoalleles in four sub-genomes of *F.* × *ananassa*, distributed on the homoeologous group of chromosomes Fvb6-1 to Fvb6-4. The chromosome Fvb6 of *F. vesca* also harbors all clade II NPR proteins encoding genes except for *FveNPR31*. The homoeologous chromosome group Fvb6-1~Fv6-4 in octoploid strawberry possesses 11 *NPR3*-like homoalleles similarly distributed, although the C sub-genome only has two loci. *FveNPR31*, the potential negative regulator of strawberry SA-dependent defense, locates on Fvb3 in *F. vesca*, whose homoallele is absent in the dominant A sub-genome [14]. *FveNPR31* homoalleles occur on the homoeologous chromosome group Fvb3-1~Fv3-3 in B, C, and D sub-genomes.

### 2.4. Cis-Elements in the Promoter Regions of NPR-like Genes from F. vesca and F. × ananassa

Characterizing the occurrence of cis-elements in gene promoter region will facilitate our understanding of its regulatory expression. The 2500-bp upstream promoter sequences of *NPR*-like genes in both woodland and octoploid strawberries were selected and analyzed using PlantCare (Figure 5). The elements identified were sorted into three types, including hormone response, stress response, and others. Information for the sequences and functional categories of identified cis-elements was listed in Appendix A. It is worth noting that in the current prediction from PlantCare (Figure 5, Appendix A), the TC-rich repeats (yellow block) may be responsible for the defense and stress responses [40], which was often overlapped with the MYB binding site. The WBOXATNPR1 (TTGAC) (red block) was a fungal elicitor element identical to the binding site for WRKY involved in plant defense response [41,42]. The GT1/TCA-element (red circle) is a salicylic acid responsive element [43,44]. The activation sequence-1-like (as-1) (yellow circle with black line) interpreted as auxin and salicylic acid responsive was indeed identical to the methyl-jasmonate (MeJA) responsive element, which was composed of two CGTCA or TGACG-motifs in close proximity, making up the binding site for TGA, involved in plant responses to auxins; salicylic acid (SA); MeJA; and further stressors, including hydrogen peroxide, dehydration, salt stress, etc. [45,46,47].

For *Fragaria NPR*-like genes encoding protein of clade I, the most abundant and ubiquitous cis-elements identified were MYB-binding sites. More than 10 motifs of this type were observed in promoters of *FveNPR1*, *FxaNPR1a*, *-1b*, and *-1d*. Each member of clade I possesses one or two SA-responsive elements. Over three as-1 like elements were detected in the promoters of all *NPR1*-like genes except for *FxaNPR1b*, whose promoter only contains two copies of this element. One copy of TC-rich repeats involved in the defense and stress response was observed in the promoters of *FxaNPR1a*, *-1b*, *-1d*, and *-1e* (masked by Myb-binding sites), but it was absent in the promoters of *FveNPR1* and *FxaNPR1c*. Surprisingly, the WBOXATNPR1 (W-box, TTGAC) motif (red block) was completely absent in the promoters of all *Fragaria NPR1*-like genes. The hypothetical model might not be suitable for strawberry NPR1, whose transcription factors WRKY6 and WRKY18 associate with the NPR1 promoter through the W-box motif for the transcriptional regulation of *NPR1* gene in *Arabidopsis* [48].

Three subgroups of clade II *NPR*-like genes varied in terms of the composition of cis-regulatory elements in their promoter regions. One copy of W-box was only present in the upstream of *FveNPR31* and *FxaNPR31a*, although all four *NPR31*-like genes possess as-1 like and SA-responsive elements in their promoter regions. However, *FveNPR32a* and its three homoalleles *FxaNPR32a*, *-32b*, and *-32d* similarly had two copies of W-box in their promoters close to translation starting code. By contrast, *FveNPR33* and its three homoalleles *FxaNPR33a*, *-33b*, and *-33c* uniformly had two copies of W-box far away from the translation starting code. Generally, promoter regions of *FveNPR33* and its homoalleles contained more SA-responsive elements (3–6 copies) than the remaining members (0–2 copies) in clade II. In addition, *FxaNPR32c* was the unique member in the strawberry *NPR*-like family whose promoter did not have an SA-responsive element.

Among *NPR*-like genes of clade III, *FveNPR5* was the unique one possessing a W-box motif in its promoter region, while its four homoalleles in *F.* × *ananassa* did not have that motif. The promoters of all genes in this clade contained as-1 like and SA-responsive elements. Notably, there were as many as eight SA-responsive elements in the *FxaNPR5d* promoter.

### 2.5. Transcriptional Responses of Strawberry NPR-like Genes to C. fructicola

To reveal whether and how strawberry motivates its NPR-like family members in responses to biotic stress, strawberries, including *F. vesca* ‘Hawaii4’, *F.* × *ananassa* cultivars ‘Camarosa’, and ‘Benihoppe’, were inoculated with *C. fructicola* (strain CGMCC3.17371) of *C. gloeosporioides* complex, a dominant species causing strawberry anthracnose crown rot worldwide. Whole plants were spray-inoculated, and clear necrosis lesions appeared in ‘Benihoppe’ at 2 days post inoculation (dpi), while in ‘Camarosa’ and ‘Hawaii4’, small lesions only appeared at 4 dpi (data omitted). The detached leaves were inoculated on wounded sites for phenotyping (Figure 6A). Significantly larger lesions were observed on leaves of ‘Benihoppe’ than those of ‘Hawaii4’ and ‘Camarosa’ since 4 dpi, which was consistent with the field observation that ‘Benihoppe’, is more susceptible than most other genotypes to *C. fructicola*.

Expression responses of strawberry *NPR*-like family genes were investigated in RNA samples from the leaves of whole plants spray-inoculated with *C. fructicola* or mock-treated. First, semi-quantitative RT-PCR was carried out (Figure 6B). The results showed that the transcripts of whole *NPR*-like family genes could be detected in ‘Hawaii4’ leaves, although *FveNPR1* and *FveNPR5* were expressed at much lower levels than *FveNPR31-33*. Indeed, five-time more cDNA templates combined with 40-cycle repeats in PCR settings were required for detecting the transcripts of *FveNPR1* and *-5*. The same primerset matching *FveNPR5* was used for analysis in octoploid strawberry. It was revealed that the total expression of several *FxaNPR5s* alleles was significantly higher than that of a single *FveNPR5* in ‘Hawaii4’. As compared with mock treatment, there was a mild pathogen invasion-induced expression of *FxaNPR5s* at 6 hpi in ‘Benihoppe’.

Concerning the potential central regulatory importance of clade I *NPR*-like genes in plant immunity, allele-specific primers were designed for expression analysis in octoploid strawberry. *FxaNPR1a* expression was always detected in ‘Camarosa’ leaves either infected or mock-treated, and a pathogen-induced transcription decrease in this allele was observed at 0 (about 15 min or less post inoculation), 6, 24, and 48 hpi. However, *FxaNPR1a* expression was only occasionally detected in ‘Benihoppe’ at a lower level, although at even lower levels, expression of the remaining three alleles *FxaNPR1b-d* was detected in ‘Camarosa’ while scarcely in ‘Benihoppe’. The transcription of *FxaNPR1e* was never detected in the current work. Variations in the expression patterns of *FxaNPR1a-d* observed in two cultivars might be caused by different genetic backgrounds and less primer efficiency influenced by potential single nucleotide polymorphisms in ‘Benihoppe’.

Indeed, the expression of *FveNPR1*, *-5*, and their homoalleles in octoploid strawberry was too weak to be investigated via quantitative RT-PCR (qPCR). Accordingly, qPCR analysis was only performed for *FveNPR31-33* in ‘Hawaii4’ and their homoalleles in ‘Benihoppe’ (Figure 7). It was revealed that *FveNPR31* expression was quickly suppressed upon inoculation at 0 hpi (less than 15 min after inoculation) and significantly induced at 96 hpi in ‘Hawaii4’. Similarly, the total expression of *FxaNPR31abc* was up-regulated at 48 hpi (significantly) and 96 hpi in ‘Benihoppe’. The expression of *FveNPR32a*, *-32b*, and *-33* was not significantly altered in ‘Hawaii4’ infected with *C. fructicola*, while the total transcription of their homoalleles in ‘Benihoppe’ was significantly suppressed at certain stages post inoculation. In addition, a clear induction of the stress-responsive defense gene *FvePR10a* and its homoalleles *FxaPR10a* was detected in both ‘Hawaii4’ and ‘Benihoppe’ at 48 hpi and 96 hpi. The well-known SA-pathway marker gene *FvePR1* was suppressed upon *C. fructicola* infection in two strawberries (data omitted).

### 2.6. Transcriptome Profile of NPR-like and Related Genes in Strawberry Infected with C. fructicola

The pivotal role of NPR-mediated defense response lies in the interaction with other proteins, including many transcription factors, since NPR1/3/4 are transcription cofactors without a DNA binding domain [6]. Potential-interacting proteins were sought for FveNPRs at STRING (https//cn.string-db.org, accessed on 26 February 2022). Accordingly, a protein–protein interaction network consisting of 65 modes and 371 edges was generated largely based on the known interactions from curated databases and experimentally determined interactions (Figure 8).

Following functional enrichment analysis, it was revealed that FveNPR1 and FveNPR31, the potential shuttles between nuclear and cytoplasm predicted in PSORT, might directly interact with four TGA transcriptional factors for gene expression regulation, with one NIMIN-2 and two CUL3a for regulating protein stability, and with MAPK for signaling magnification. Furthermore, two CUL3a proteins could bridge FveNPR1 and -31 with three Ring-box 1a-like proteins, BPM (adapter protein containing BTB/POZ and MATH domains that link cell surface receptors and downstream kinase cascades), and CSN (COP9 signalosome complex subunit like). Later, two UBE (Ubiquitin-conjugating enzyme E27-like), three BTB/POZ domain-containing proteins, and 26S proteasome were recruited into this FveNPR1/31-centored network bridged by CUL3a. In another direction, MAPK might link FveNPR1 and -31 with NDPK (nucleoside diphosphate kinase 2, chloroplastic-like), additional MAPK, and PP2C (protein phosphatase 2C and cyclic nucleotide-binding/kinase). MAPK and PP2C further linked FveNPR1/31 with PTP (protein–tyrosine–phosphatase), GNB (guanine nucleotide-binding protein), LRR-RLK (leucine-rich repeat receptor-like serine/threonine/tyrosine-protein kinase), WRKY, and HY5-like transcription factors.

Since transcriptome profiling could provide a comprehensive view of a certain regulatory network at a transcription level, we revisited previous RNA sequencing (RNA-seq) data in the current lab to determine how strawberry activates related defenses against the invasion of *C. fructicola* via motivating *NPR*-like and related genes in a susceptible cv. ‘Jiuxiang’ [35]. In control strawberry (mature healthy ‘Jiuxiang’ mock-treated with water), the relative levels of *FveNPRs* transcripts ranked as follows: *FveNPR32* > *FveNPR33* > *FveNPR31* > *FveNPR5* > *FveNPR1* (Figure 9A). *C. fructicola* infection significantly induced the expression of *FveNPR31*, moderately upregulated *FveNPR32* and *-33*, and inhibited *FveNPR5* in ‘Jiuxiang’ (Figure 9A). *FveNPR1* expression was extremely weak, with minor fluctuations. The expression profile of *FveNPR31* revealed in RNA-seq was congruent with our RT-PCR analysis in ‘Camarosa’ and ‘Benihoppe’. A set of potential interacting protein (Figure 8) coding genes were revealed to be differently expressed during invasion with *C. fructicola*. The transcription of genes encoding the remaining proteins in that PPI network were not differently altered in response to *C. fructicola* infection (Appendix A). In addition, the expression profiles of 14 genes involved in SA-related signaling were identified (Figure 9B). Upon *C. fructicola* infection, strawberry SA biosynthesis-related genes *ICS* and *PALs* were differentially regulated. The SA pathway marker genes *PR1* and *PR10b* were suppressed, while *PR10a* was upregulated, largely matching the current RT-PCR analysis.

## 3. Discussion

Generally, *NPR*-like genes belong to a small family that is highly conserved in many plants. *Arabidopsis* harbors six *NPR*-like genes [50]. There are five *NPR*-like members in rice [51] and *Persea americana* [13]. Notably, the genomes of Chinese pear (*Pyrus bretschneideri* Rehd, cv. Dangshan Suli) and apple (*Malus* × *domestica*) consist of nine and eight *NPR*-like members, respectively [20]. Tetraploid emmer wheat and hexaploidy wheat embrace 12 and 17 *NPR*-like genes, respectively [15]. The allotetraploid genomes of oilseed rape (*Brassica napus*) [52] and mustard (*Brassica juncea*) [19] consistently harbor 19 *NPR*-like genes. In this study, a total of 23 *NPR*-like members were identified in octoploid commercial strawberry *F.* × *ananassa*. With the exception of *F. nipponica*, five to six *NPR*-like genes were identified in the genomes of diploid *Fragaria* species, including *F. vesca* (6), *F. viridis* (6), *F. iinumae* (5), and *F. nilgerrensis* (5). The *NPR*-like family in *F. vesca* genome has been previously reported [14], which was largely consistent with the current work, except for that one more member was recruited due to the division of gene28770 (previous *FvNPRL-3* in strawberry v1.0-hybrid version) into FvH4_6g38830 (*FveNPR32a*) and FvH4_6g38821 (*FveNPR32b*), which might have resulted from alternative splicing. Apart from *FveNPR32a* and *FveNPR32b* in *F. vesca*, there were an additional three pairs of genes physically linked, which might be potentially one locus with alternative splicing patterns. These were *FvirNPR32a* and *FvirNPR32b* in *F. viridis*, *FxaNPR32a* and *FxaNPR32e*, as well as *FxaNPR32b* and *FxaNPR32g* in *F.* × *ananassa*. Moreover, it is worthwhile to note that current *FveNPR1* corresponds to the FvH4_1g07810 (gene12668), previously referred to as *FvNPR1* [53] or *FvNPRL-4* [14]. Current *FveNPR31* is virtually the FvH4_3g11950 (gene20070), previously referred to as *FvNPR31* [53], *FvNPRL-1* [14], or *FvNPR1* [35,36]. Furthermore, the primers for detecting *FaNPR1* expression previously [37] fit with all four alleles of *FxaNPR33a-d* in the current work.

Phylogenetic analysis unambiguously clustered *Fragaria NPR*-like genes into three clades as in all other plants reported. Among NPRs of a certain clade from diverse plants, the conservation is widely observed in the relative length of the exons; the exon–intron organization; and the functionally important residues, motifs, and domains. Interestingly, the pI values of NPRs belonging to a certain clade might be an exceptional case, not conserved among diverse plants. For example, *Arabidopsis* NPR1 protein has a pI value of 5.86 (www.arabidopsis.org), and the pI values of clade I NPR-likes in three wheat species with distinct chromosome ploidy ranged from 5.18 to 5.88 [15], while in six strawberry species this parameter varied between 6.17 and 6.74. *NPR*-like genes in *Fragaria* species might have a unique evolutionary history after the divergence from *Arabidopsis*. More structural evidence supported that NPR-like proteins in strawberry might have functional divergence or a function somewhat partially different from their orthologs in *Arabidopsis*.

MEME analysis revealed the presence of additional motifs before the BTB/POZ domain in strawberry NPRs of clade II and III (Appendix A), which were not observed in *Arabidopsis*. Sequences of potential IkB-like phosphodegron in clade I NPR-like proteins were not completely conserved in *Arabidopsis* and strawberry (Appendix A). The Cysteine residue C156 in AtNPR1 related to facilitating NPR1 oligomerization was substituted by Alanine (*F. nilgerrensis*) or Serine (the rest five strawberry species) residue (Appendix A) [54]. A subgroup represented by *FveNPR32a* in clade II similarly was composed of three exons, while the other members of clade II consistently have four exons like *Arabidopsis* orthologues. In addition, the deduced amino acid sequences of *FveNPR32a* and its subgroup members have a variant form of the putative hinge region ‘FENRV’ (SA binding pocket) in the C terminal, while all *Fragaria* members of clade I except for FiinNPR1 (‘LENR’ followed by a deletion of three amino acids) and the other members of clade II uniformly harbor ‘LENRV’ (Appendix A), which was identical to that in AtNPR1 [39]. Strikingly, the putative hinge region in AtNPR3/4 is composed of ‘LEKRV’, which was not observed in *Fragaria* spp. Previous work suggested that AtNPR4 has an affinity with SA five times higher than that of AtNPR3 and AtNPR1 [4,55]. Apparently, the SA-binding core (SBC) is wider than the hinge region [9].

In contrast to the NPR-like members of clade III, which are mainly involved in leaf and flower development [56,57], NPR family members of clade I and II play broad roles in plant immunity [6,7,58]. The degradation of the transcription cofactor nonexpresser of *PR* genes 1 (NPR1) acts as a molecular switch in many plants, and NPR4 is indispensable for the function of NPR1 [6]. Previously, the transcription of *FveNPR1*, the *NPR1*-like gene (gene12668, FvH4_1g07810, and *FvNPRL-4*), had never been detected in strawberry except for in an RNA sequencing analysis [14,34,35,36,53]. RNAseq in early-stage fruits and achenes of woodland strawberry [59] revealed *FveNPR1* was largely considered as non-expressed, with transcripts having average reads per kilobase per million (RPKM) lower than 0.3 in all reproductive and vegetative organs except for a peak in achene ghost at stage 5 (10–13 days post anthesis, RPKM: 6.79) and achene wall at stage 1–4 (RPKM: 0.45–3.04). In the current work, based on a comprehensive phylogeny analysis for NPR-like family in octoploid and diploid strawberry, allele specific primers were available for the RT-PCR analysis of *NPR1*-like genes in strawberry. The expression of *FveNPR1* and one allele *FxaNPR1a* (FxaC_1g08070) was clearly detected in the mature leaves of ‘Hawaii4’ and ‘Camarosa’ infected or not infected with *C. fructicola* (Figure 6B). A fast and obvious inhibition effect of pathogen inoculation was observed for *FxaNPR1a* transcription, and a similar but weak tendency for *FveNPR1* was also detected. Still, its transcript level was too weak to be quantitatively analyzed. Reasonable explanations would be that NPR1 is strictly regulated at both transcription and protein levels, together with a highly efficient translation system for this master regulator. Virtually, it has been suggested that upon infection with pathogens, the translational induction of resistance-related protein may precede transcriptional alteration in plants [60]. Together, both molecular characterization and expression analysis indicated that FveNPR1 and FxaNPR1a were active in strawberry and could respond to *C. fructicola* infection at a transcriptional level, implying that they probably function in strawberry like their orthologue NPR1 in *Arabidopsis*.

Among NPR-like members of clade II in strawberry, FveNPR31 has been functionally studied [14,53]. Although RNA interference and the overexpression of *FveNPR3* in ‘Camarosa’ strawberry did not result in decisive conclusions on its role, the overexpression of this gene in wild-type and mutant (*npr1-1*, *npr3-1*, and *npr4-3*) *Arabidopsis* strongly evidenced that it might function as a negative regulator of plant immunity such as AtNPR3 or AtNPR4 in *Arabidopsis* [53]. The transcription of *FveNPR31* and its homoalleles could be induced by exogenous SA and *Colletotrichum* spp., or suppressed by exogenous MeJA and *Podosphaera aphanis*, dynamically varying with the cultivar genotype [14,34,35,36,37,53]. Based on our previous [35,36] and the current work under the same inoculation conditions with the same *C. fructicola* strain, it was concluded that in a susceptible cultivar such as ‘Jiuxiang’ or ‘Camarosa’, the transcription of *FxaNPR31s* could be quickly motivated upon *C. fructicola* inoculation but soon relaxed at 6 hpi and again up-regulated in the later stage when the pathogens experienced a necrotrophic way for colonization (simultaneously with necrosis syndromes visible). All these observations triggered speculation that the transcription of strawberry *NPR31*, the potential negative regulator of SA-dependent immunity, might be hijacked by *Colletotrichum* pathogen at an early stage (1–2 hpi), but it was soon quenched (6 hpi) when strawberry begun to frustrate this invasion via motivating multiple defense strategies such as PTI, ETI, and SAR. Furthermore, when necrosis syndrome appeared and the hemibiotrophic pathogen *Colletotrichum* initiated a necrotrophic life (2 dpi and later), there was a transition of defense strategy in strawberry to largely rely on the JA-mediated network, while SA-mediated pathways might be restricted through the activation of the negative regulator *FveNPR31*.

Finally, to summarize the transcriptional changes observed in both RNAseq and RT-PCR for strawberry *NPR1*, *NPR31*, and related partners suggested from the PPI prediction, schematic representations were proposed to facilitate our understanding of molecular mechanisms underlying NPR-related immunity during strawberry-*Colletotrichum* spp. interaction (Figure 10A,B). A glimpse of these transcriptional events would shed some light on how strawberry orchestrates NPRs-centered defense system against *C. fructicola*.

## 4. Materials and Methods

### 4.1. Plant Materials, Growth Conditions, and Inoculation with C. fructicola

The woodland strawberry *Fragaria vesca* ‘Hawaii4’ and cultivated strawberry *F.* × *ananassa* cultivars ‘Camarosa’ and ‘Benihoppe’ were used. Healthy plants with 6–8 compound leaves were developed from stolon-derived seedlings after three months’ cultivation in a greenhouse. These plants were transplanted into pots and cultivated in a growth chamber (GXZ-1000, Jiangnan Instrument Factory, Ningbo, China) under a 12 h-light/12 h-dark cycle at constant 25 °C for two more weeks before inoculation. *C. fructicola* strain CGMCC3.17371 was used for preparing inoculum. Inoculation was carried out in two ways [61]. For RNA extraction and gene expression analysis, whole plants were spray-inoculated, largely as previously described [35]. For ‘Hawaii4’ and ‘Benihoppe’, 18 plants divided into three groups were sprayed with *C. fructicola* conidial suspensions at a concentration of 2 × 10^6^ per ml with 0.01% (*v*/*v*) Tween-20. Simultaneously, additional 9 plants were mock-treated with sterile ddH_2_O containing 0.01% (*v*/*v*) Tween-20. For ‘Camarosa’, 12 plants were inoculated and 6 plants were mock-treated. The third to fifth fully expanded leaves from three (mock-treated) or six plants (*C. fructicola* inoculated) were harvested and pooled for one replicate at 0, 6, 24, 48, and 96 hpi. A total of three biological repeats were obtained for each treatment × time combination. For 0 hpi, sampling was accomplished within 15 min after spaying. Leaf blades were immediately frozen in liquid nitrogen and then stored at −80 °C before RNA purification.

In addition, in-vitro inoculation on wounded detached leaves was performed as previously described with minor modifications [62]. The third or fourth fully expanded leaves were detached with the cut end wrapped with paraffin ceresin and placed on wet sterile filter paper in sealed petri dishes (diameter, 15 cm) to maintain humidity. Each leaf blade was injured with a pipette tip at four sites. Two 10-μL drops of freshly prepared conidia solutions at a concentration of 2 × 10^6^ per mL with 0.01% (*v*/*v*) Tween 20 were deposited on the injured sites on the right side, with mock treatment (sterile water with 0.01% (*v*/*v*) Tween 20) on the left side. A total of nine leaf blades were inoculated for each strawberry variety. The typical necrosis of anthracnose syndromes was photographed at 4dpi and 5 dpi.

### 4.2. Identification and Phylogenic Analysis of NPR-like Genes in Fragaria Species

To identify strawberry *NPR*-like genes, *F. vesca* Genome CDS (v4.0.a2) at strawberry genome database (SGD, http://www.strawberryblast.ml:8080/strawberry/viroblast.php, accessed on 11 October 2021) was searched via tblastn using AtNPRs protein sequences from https://www.arabidopsis.org/, acessed on 28 February 2022, as queries [25]. All candidates were further validated through PFAM (http://pfam.xfam.org/, acessed on 28 February 2022) and CDD analysis (www.ncbi.nlm.nih.gov/Structure/cdd/, acessed on 28 February 2022). A similar process for the identification of NPR-like loci in *F. iinumae*, *F. nilgerrensis*, *F.viridis*, and *F. nipponica* genomes was carried out against databases (*F. iinume* Genome V1.0 CDS, *F. nilgerrensis* Genome V1.0 CDS, *F. viridis* SCBG Genome V1.0 transcripts, and *F. nipponica* Genome V1.0 (FNI-r1.1) scaffolds) at https://www.rosaceae.org/, acessed on 28 February 2022, [63]. Additional HMM-based gene prediction for NPR-like candidates from *F. nipponica* was performed at Softberry (http://linux1.softberry.com/berry.phtml?topic=fgenesh&group=programs&subgroup=gfind, accessed on 11 October 2021). Putative *NPR*-like genes in octoploid strawberry were later identified through searching against the masked version of the *F.* × *ananassa* Genome CDS (v1.0.a2) at SGD using all six *FveNPR-like* members. PFAM combined with CDD analysis was again carried out for validating all candidates.

NPR-like proteins in six *Fragaria* species were symbolized with specific numbers followed by FxaC, FvH4, evm.model.scaf, FnYN, evm.model.ctg, and FNI, representing those from *F.* × *ananassa*, *F. vesca*, *F. iinume*, *F. nilgerrensis*, *F. viridis*, and *F. nipponica*, respectively. Phylogenetic analysis was performed for a set of 54 NPR-like proteins from *Arabidopsis* and aforementioned six *Fragaria* species using a neighbor-joining (NJ) statistical method in MEGA v7.0 [64]. The corresponding tree was constructed using 1000 re-samplings in MEGA and visualized via using the interactive Tree Of Life (iTOL) [65].

The in silico prediction of biochemical characteristics was performed for each strawberry NPR-like member. The subcellular localization was predicted at https://wolfpsort.hgc.jp/, accessed on 12 October 2021. The isoelectric point (pI) and molecular weight of the protein sequences were calculated using the online Compute pI and Mw tool at https://web.expasy.org/, accessed on 12 October 2021. 

### 4.3. Exon-Intron Structure in NPR-like Genes and Domain Organization in Deduced Proteins

Information for the exon–intron structure of the *NPR*-like genes was obtained by aligning the coding sequences (CDS) from SGD or GDR with the corresponding genomic DNA sequences from GDR. A systematic analysis was performed via using the gene structure display serve (GSDS) 2.0 (http://gsds.cbi.pku.edu.cn/, accessed on 27 February 2022). A Newick-type file was generated in MEGA7.0 with the CDS nucleotide sequences of all *Fragaria NPR*-like genes after MUSCLE clustering. This file was further introduced into GSDS as a neighbor-joining tree to arrange the exon–intron structures of strawberry *NPR*-like genes.

Conserved protein domains were identified at PFAM and illustrated with IBS1.0 (Illustrator for Biological Sequences 1.0) [66]. To identify the conserved motifs in strawberry NPR proteins, the MEME program at http://meme-suite.org/ (accessed on 11 October 2021) was employed using the following settings: optimum width, 15–60; number of repetitions, any; and maximum number of motifs, 10. Multiple sequence alignment was performed via using Clustal Omega at https://www.ebi.ac.uk/Tools/msa/clustal/ (accessed on 21 December 2021) and further visualized using Jalview (https://www.jalview.org/, downloaded on 25 September 2021) [67]. The percentage of identical amino acids among strawberry NPR-like proteins was calculated after ClustalW analysis.

### 4.4. Chromosomal Locations of NPR-like Genes in F. vesca and F. × ananassa

The genomic locations of *NPR*-like genes were obtained from *F. vesca* and *F.* × *ananassa* genome databases at GDR [23] and assigned to distinct chromosomes or subgenomes [29]. Mapchart1.0. software was used to draw the physical maps showing the distribution of *NPR*-like genes [68].

### 4.5. Cis-Elements in Promoters of NPR-like Loci in F. vesca and F. × ananassa

The promoter sequence of 2500-bp in length was extracted from GDR at the upstream of the ATG translation start codon for each *NPR*-like genes in *F. vesca* and *F.* × *ananassa*. Cis-elements were predicted in the PlantCare database at http://bioinformatics.psb.ugent.be/webtools/plantcare/html (accessed on 9 February 2022) [69] and illustrated via using TBtools package [70].

### 4.6. Protein-Protein Interaction Prediction for Strawberry NPR-like Proteins

Potential protein–protein interactions (PPI) were predicted for the diploid strawberry NPR-like proteins from *F. vesca* using the STRING database version 11.0 (http://string-db.org/cgi/input.pl, accessed on 26 February 2022) [71]. Functional enrichment analysis started with all FveNPR members beyond FveNPR5 and resulted in a network harboring 65 nodes linked with 165 expected edges (expected interactions) and a total of 371 edges (potential interactions included) with a *p*-value less than 1.0 × 10^−16^. The interaction network was built via seven times of PPI enrichment analysis, resulting in ever increasing proteins of known interactions from curated databases and experimentally determined, predicted interactions due to the gene neighborhood, gene fusions, and gene co-occurrence, as well as other potential interactions based on text-mining, co-expression, and protein homology.

### 4.7. RNA Purification, RT-PCR, and RNA Sequencing

The infected or mock-treated plant materials were homogenized by liquid nitrogen, and total RNA was extracted using the OMEGA Plant RNA kit (OMEGA Bio-tek, Inc. Cat#R6827, Norcross, GA, USA) no longer than two weeks after sampling. For RT-PCR, cDNAs synthesis was achieved by using the HiScript III RT SuperMix for qPCR kit with gDNA wiper (Vazyme, Lot#R323, Nanjing, China).

RT-PCR primers matching each *FveNPR* and its homoeologous *FxaNPRs* were first designed based on sequence alignment to avoid single nucleotide polymorphisms. Furthermore, allele-specific primers were designed for clade I *NPR1*-like loci in octoploid strawberry. The information for all primers was shown in Appendix A. For *NPR*-like family genes in *F. vesca* and *NPR1*-like genes in *F.* × *ananassa*, gene-specific primer pairs were used to distinguish each allele. For *NPR*-like genes of clade II and III, the SA signaling marker gene *PR10a*, and thge two reference genes *EF1α* (FvH4_3g33150, gene28639) and *GAPDH2* (FvH4_4g24420, gene07104) [49], the same primer set was used for analysis in *F. vesca* and *F.* × *ananassa*, which means that the homologous alleles were not distinguished in octoploid strawberry.

Semi-quantitative RT-PCR was accomplished in an ETC 821 DNA amplifier (Eastwin, Beijing, China) via using the 2xTaq PCR Master Mix (Cat #PM1101, Biosune, Shanghai, China) in a 20-μL cocktail. PCR cycles for the internal control *EF1α* and *NPR*-likes were 28 and 40, respectively. For *EF1α* and strawberry *NPR3*-like genes of clade II, 0.2-μL original cDNAs was used as a template, while for *NPR*-like genes of clade I and III, 1-μL original cDNAs was used. After thermal cycles, 6.5 or 7.5-μL PCR products were detected on 2% agarose gel. Three biological replicates were tested.

Real-time quantitative PCR (qPCR) assay was performed for *NPR3*-like genes of clade II in *F. vesca* ‘Hawaii4’ and *F.* × *ananassa* cv. ‘Benihoppe’. The ChamQ^TM^ Universal SYBR qPCR Master Mix (Vazyme, Lot#Q711, Nanjing, China) was used in a 12-μL cocktail following user manuals. qPCR was conducted on a Light Cycler 480 (Roche, USA). For all *NPR3*-like genes, primer amplification efficiency (E) was first examined with a gradient of diluted cDNA templates. *FveEF1a* and *FveGAPDH2* were used for expression normalization. qPCR analysis was performed using the (1 + E)^−∆∆Ct^ method combined with a normalization factor calibrated with geNorm software [72]. Three biological replicates were examined with each in three technical repeats. Student *t*-test analysis was performed to reveal the significance of differences.

RNA sequencing for *C. fructicola*-infected strawberry cv. ‘Jiuxiang’ in the current laboratory has been reported previously [35].

## 5. Conclusions

In summary, this study reports a comprehensive identification of *NPR*-like family genes in the genomes of octoploid strawberry and its five diploid relatives based on updated whole genome information. Phylogeny analysis and molecular characterization enabled us to recognize all 23 NPR-like alleles in commercial strawberry. The mystery of putative NPR1 orthologue in strawberry may be due to the fact its transcripts in most organs are at an extremely low level and had been never detected before this work due to impaired efficiency in primers confused with distinct homoalleles. We provided experimental evidence for the view that *FveNPR1* and *FxaNPR1a* were active genes transcriptionally responsive to pathogen invasion in strawberry. Furthermore, *FveNPR31* and three homoalleles *FxaNPR31a-c* in octoploid strawberry were transiently induced (perhaps hijacked by pathogens) at an early stage and finally motivated to reduce the antagonizing effect of the SA-dependent pathway on JA-mediated resistance, which is required for plants defending a necrotrophic pathogen. Thus, FveNPR1 and FxaNPR1a, as well as FveNPR31 and FxaNPR31a-c, are putative functional orthologues of AtNPR1 and AtNPR3/4, respectively, serving as important candidate genes for future functional study aimed at improving strawberry disease resistance.

## Figures and Tables

**Figure 1 plants-11-01589-f001:**
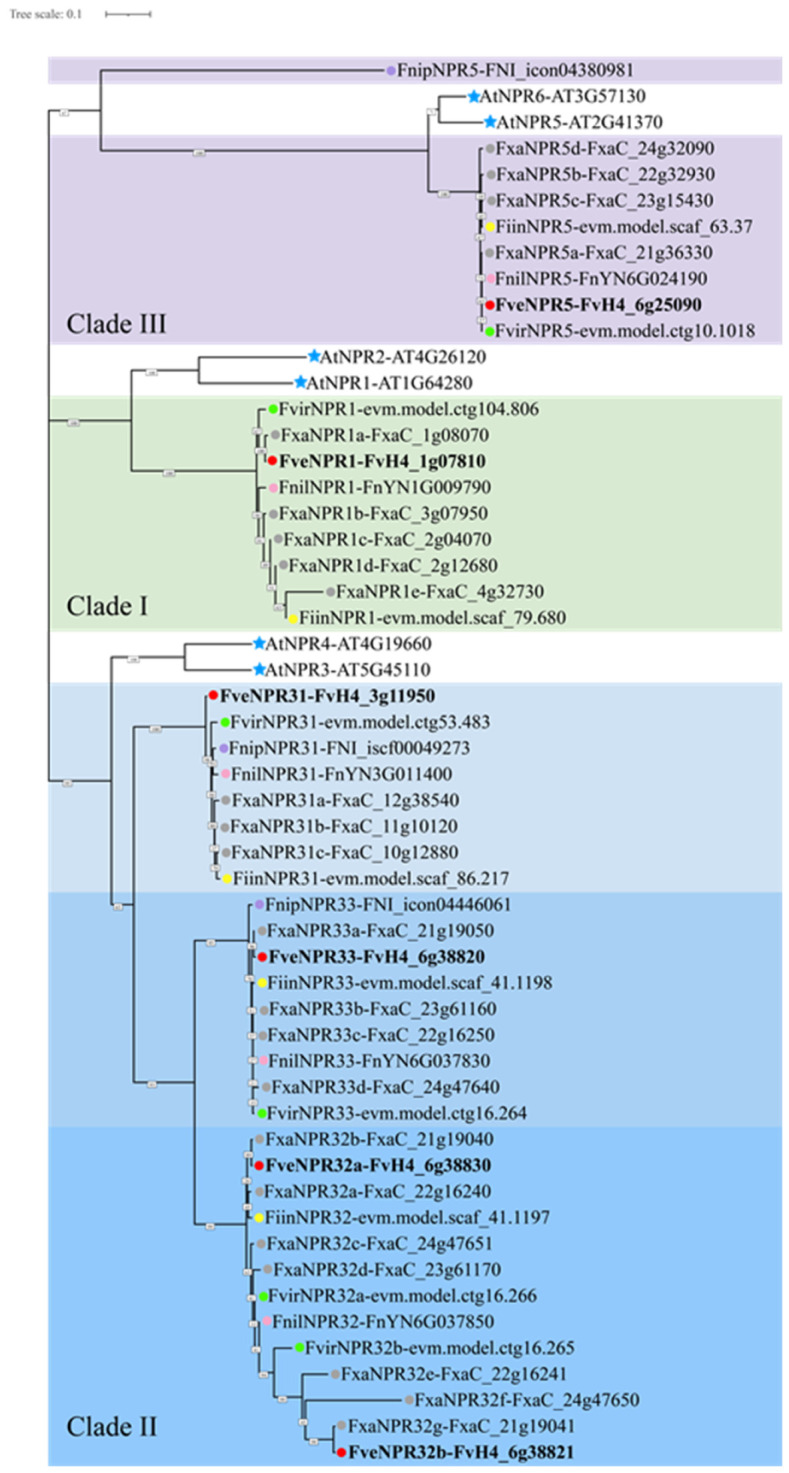
The phylogenetic relationships of NONEXPRESSOR OF PATHOGENESIS-RELATED GENES 1 (NPR1)-like proteins from *Arabidopsis thaliana*, five diploid strawberries, and the octoploid strawberry. The tree was clustered into three clades (I, II, and III) shaded with different colors. The length of branches indicates the relative phylogenetic relationship, and the bootstrap values near branches for confidence. Geometries with different shapes and colors are used to symbolize different NPR members, with a star for *Arabidopsis*, and cycles in red, yellow, green, purple, pink, and gray for *Fragaria vesca*, *F. iinumae*, *F. viridis*, *F. nipponica*, *F. nilgerrensis*, and *F.* × *ananassa*, respectively.

**Figure 2 plants-11-01589-f002:**
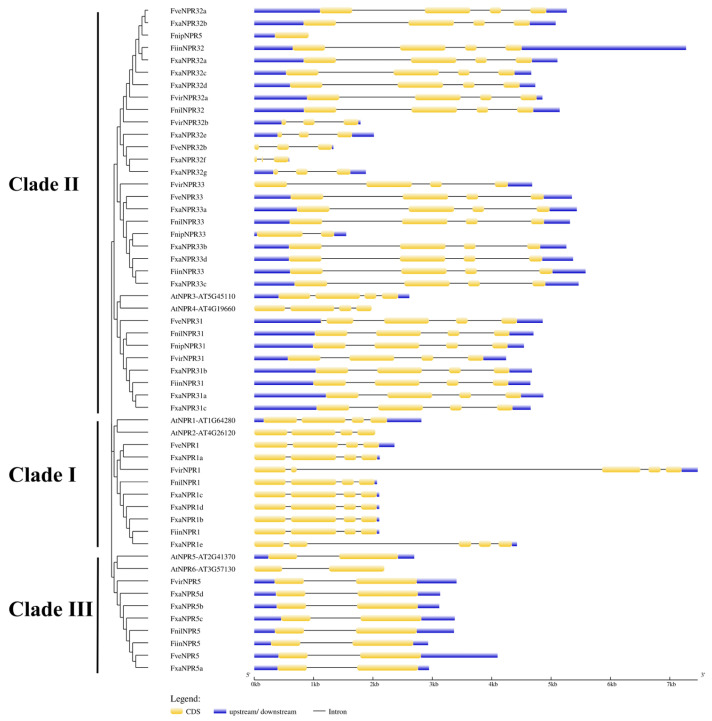
The comparative organization of exons and introns in 48 strawberry *NPR1*-like genes and 6 *AtNPRs*. The structure was produced with the gene structure display server (GSDS) at http://gsds.cbi.pku.edu.cn/Chinese.php (accessed on 27 February 2022). The blue and the yellow rectangles represent the un-translated regions (UTR) and the coding sequences (CDS), respectively. The neighbor-joining tree of the *NPR1*-like genes was generated with MUSCLE clustering by MEGA7.0 for CDS nucleotide sequences.

**Figure 3 plants-11-01589-f003:**
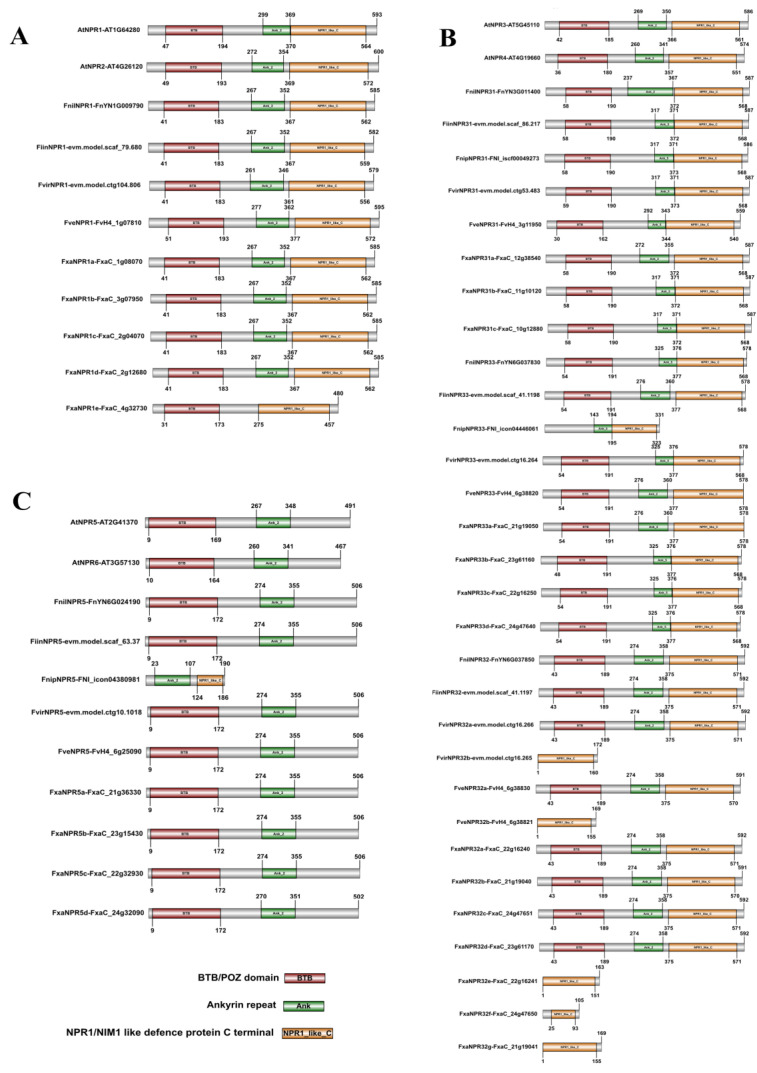
The comparative protein domain module of six AtNPRs and 48 strawberry NPR1-like sequences. The deduced protein sequences in clade I (**A**), clade II (**B**), and clade III (**C**). The locations of the conserved BTB/POZ domain, the Ankyrin repeats (Ank_2 or Ank_5), and the NPR1/NIM-like defense protein C-terminal region (NPR1_like_C) were revealed via using the web CD Search Tool at NCBI.

**Figure 4 plants-11-01589-f004:**
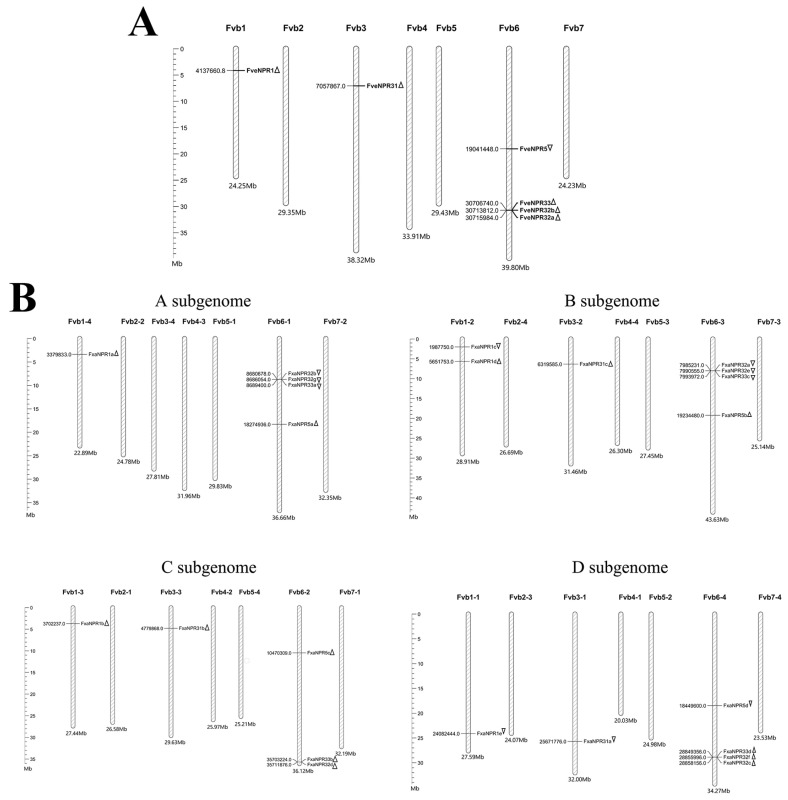
The chromosomal distributions of *NPR1*-like genes in *Fragaria vesca* (**A**) and *F.* × *ananassa* (**B**). The identity of each chromosome (of certain sub-genome) is shown at the top. The scale ruler at the left side indicates the physical distance of chromosomes in megabases (Mb). The location site and the transcriptional direction for each strawberry *NPR1*-like locus are marked as triangle.

**Figure 5 plants-11-01589-f005:**
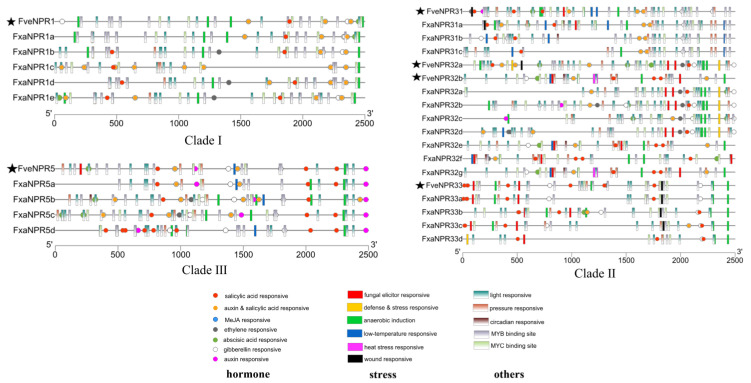
The predicted cis-elements in the promoter regions of *NPR*-like genes in *Fragaria vesca* and *F.* × *ananassa*. The promoter sequences (−2500 bp upstream of the starting code ATG) of 48 strawberry *NPR*-like genes were analyzed by PlantCare. The geometries in different colors and shapes indicate elements involved in different processes, with cycles for hormone-responsive, filled rectangles for stress-responsive, and rectangles with gradient color for the others.

**Figure 6 plants-11-01589-f006:**
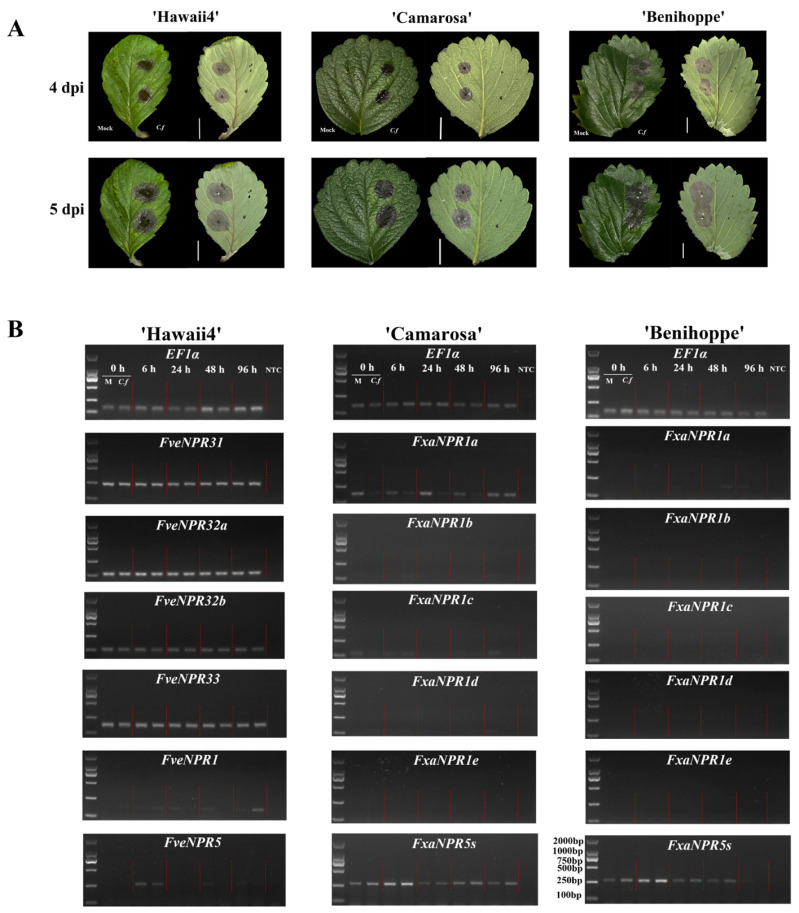
The expression responses of strawberry *NPR*-like genes upon *Colletotrichum fructicola* invasion in diploid ‘Hawaii4’ and octoploid cultivars ‘Camarosa’ and ‘Benihoppe’. (**A**) The typical symptoms induced by *C. fructicola* on wounded detached strawberry leaf blades at 25 °C were photographed at 4 and 5 dpi. The left and right side of each leaf blade (adaxial side up) was inoculated with two 10-μL droplets of sterile water with 0.01% (*v*/*v*) Tween 20 (Mock, M) and conidia suspension (2 × 10^6^ per mL, *C. f*), respectively. Scale bar, 1 cm. (**B**) The semi-quantitative RT-PCR analysis of strawberry *NPR*-like genes. The third, fourth, and fifth compound leaves of sprayed inoculated plants (*C. f*) or mock-treated (M) were harvested at different hours post inoculation (hpi). The PCR cycle numbers are 40 and 28 for *NPR1*-likes and the internal control *EF1α*, respectively. In amplification for *FveNPR31*, *-32a*, *-32b*, and *-33*, 0.2-μL original cDNAs was used as a template, while for *NPR1* and *-5* in both diploid and octoploid strawberry, 1-μL original cDNAs was used.

**Figure 7 plants-11-01589-f007:**
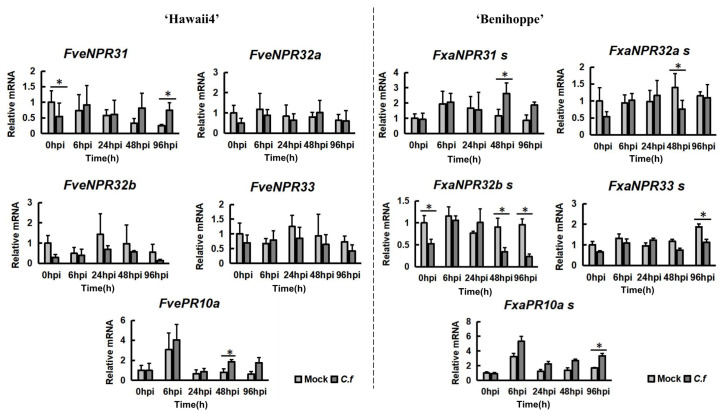
A quantitative RT-PCR analysis of strawberry *NPR3*-like genes upon *C. fructicola* invasion in diploid ‘Hawaii4’ and octoploid ‘Benihoppe’. The same RNA samples in Figure 6 were analyzed. The same primer pairs for all *NPR3*-like genes except for *NPR32b* were used in *F. vesca* ‘Hawaii4’ and *F.* × *ananassa* ‘Benihoppe’. *FvePR10a* (FvH4_4g19120) was used as a marker gene of SA-depending defense signaling pathway. The primers and corresponding target alleles amplified were shown in Appendix A. The gene name followed with ‘-s’ indicates multiple alleles detected in ‘Benihoppe’. The relative transcript levels of *NPR3*-like genes were normalized with two reference genes *EF1α* and *GAPDH2* [49] and reported as the mean of three biological replicates ± SE. The asterisks indicate significant differences based on a Student *t*-test analysis (*, *p* < 0.05).

**Figure 8 plants-11-01589-f008:**
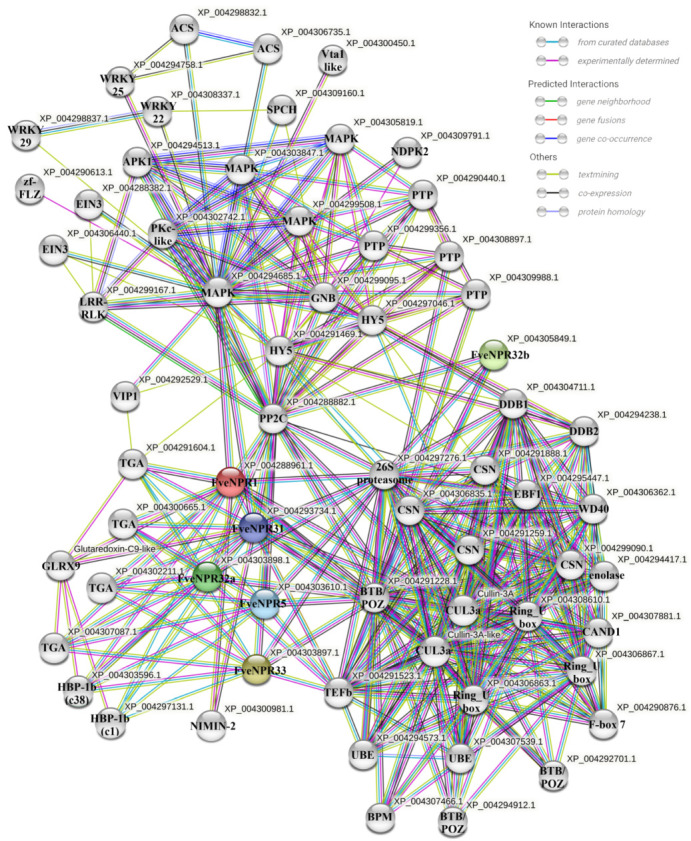
The protein–protein interaction networks proposed for the strawberry FveNPR1-like proteins in STRING (https://cn.string-db.org, accessed on 26 February 2022). The edges in different colors represent the predicted interaction relationships according to different methods. The detailed information for all potential partners is shown in Figure 9 and Appendix A.

**Figure 9 plants-11-01589-f009:**
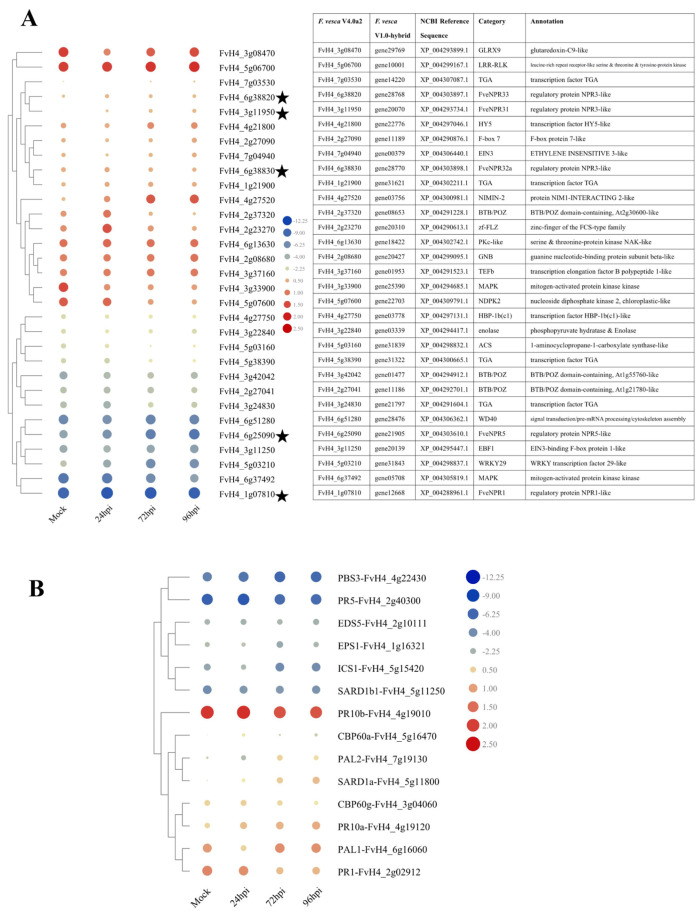
RNA-seq data showing the differential expression of genes coding proteins potentially directly or indirectly interacting with NPR1-like proteins (**A**) and the other known members in SA signaling (**B**) in strawberry during the *C. fructicola* invasion. The heatmap was generated using RPKM (reads per kilobase per million mapped reads) values normalized via Log2-transformation for each transcript in a moderate susceptible strawberry cv. ‘Jiuxiang’ mock-treated or infected by *C. fructicola* at 24, 72, or 96 hpi. The black star symbols indicate NPR family members. Strawberry materials and *C. fructicola* inoculation conditions for RNA-seq data generation have been reported previously [35].

**Figure 10 plants-11-01589-f010:**
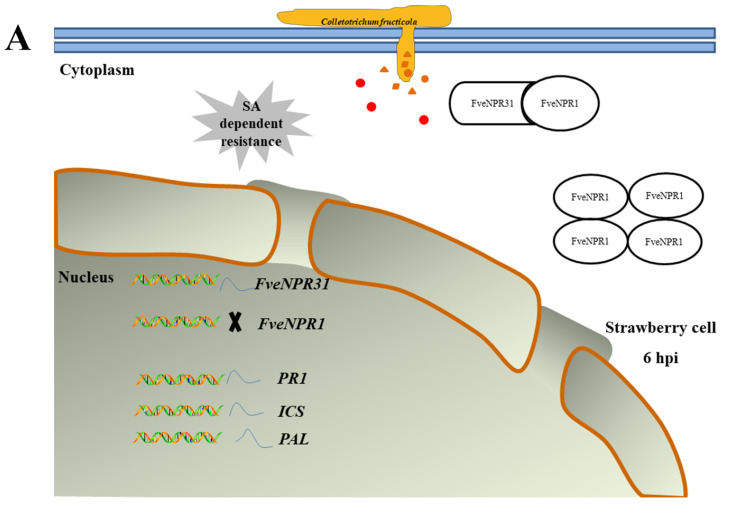
A hypothetical model depicting the transcriptional events related to NPR1- and NPR31-mediated defenses in susceptible strawberry upon invasion with a hemibiotrophic fungal pathogen *Colletotrichum fructicola*. (**A**) At 6 hpi early stage, the strawberry cell accumulates a relatively low or basal level of SA (red cycles). There is no monomeric NPR1 protein (ellipse), and it exists in cytoplasm as a large oligomer or forming heterodimer with the negative regulator NPR31. In the nucleus, the transcription of NPR1 is suppressed, while NPR31 transcription has been restored to a steady state after a transient induction. Simultaneously, the expression of pathogenesis-related gene *PR1*, as well as SA synthesis genes *ICS* and *PAL*, show a similar dynamic pattern to that of *NPR31*. All events indicate a fast quenching of SA-dependent resistance, resulting in effector (small brown shapes)-triggered susceptibility (ETS). (**B**) At the 96 hpi late necrotrophic stage, the strawberry cell contains elevated SA due to the increased expression of PAL1/2. SA-dependent resistance is partially activated after NPR1 binds SA. Enhanced NPR31 negatively regulates SA-related immunity. Meanwhile, SA-dependent resistance is antagonized by the disinhibition of MeJA-mediated defense and the activation of ethylene-related defense, which is beneficial for strawberry defending the pathogen with a necrotrophic life at 96 hpi. There exists monomeric NPR1 and NPR31 in the nucleus, which directly target TGA transcriptional factors to regulate the expression of *PR* genes. Members are in red characters for up-regulated and in blue for down-regulated.

**Table 1 plants-11-01589-t001:** The genomic and biochemical information for *NPR* genes identified in diploid and octoploid strawberries.

GENE NAME	Locus ID	Chromosomal Location	DNA (bp)	mRNA (bp)	Protein (aa)	LOC ^a^	MW ^b^ (kDa)	Pi ^c^
*FnilNPR1*	FnYN1G009790	GWHABKC00000007:1g6682576-6684604	2029	1758	585	Nuc:Ct_Nuc (6.5:6)	64.7	6.74
*FnilNPR31*	FnYN3G011400	GWHABKC00000002:3g8012487-8017100	4614	3096	587	Ct:ER (4:3)	65.3	6.39
*FnilNPR32*	FnYN6G037850	GWHABKC00000001:6g39083311-39088355	5045	2958	592	Ct	66.3	5.87
*FnilNPR33*	FnYN6G037830	GWHABKC00000001:6g39072614-39077826	5213	2660	578	Nuc	64.7	5.77
*FnilNPR5*	FnYN6G024190	GWHABKC00000001:6g24090462-24093763	3302	2430	506	CL:ER(4:3)	55.1	6.30
*FiinNPR1*	evm.model.scaf_79.680	Chr1:3535351-3537414	2064	1749	582	Ct:Ct_Nuc (6.5:6.5)	64.6	6.34
*FiinNPR31*	evm.model.scaf_86.217	Chr3:6321505-6326064	4560	3037	587	CL:Ct:ER(3:3:3)	65.3	6.15
*FiinNPR32*	evm.model.scaf_41.1197	Chr6:34174583-34181716	7134	5053	592	Nuc	66.1	5.88
*FiinNPR33*	evm.model.scaf_41.1198	Chr6:34168193-34173663	5471	2786	578	Nuc	64.8	5.71
*FiinNPR5*	evm.model.scaf_63.37	Chr6:22982676-22985547	2872	1994	506	CL:ER(4:3)	55.2	6.25
*FnipNPR31*	FNI_iscf00049273	FNI_iscf00049273.1:1606-6149	4544	3026	586	CL:Ct:ER(3:3:3)	65.2	6.48
*FnipNPR33*	FNI_icon04446061	FNI_icon04446061.1:1..1551	1551	1244	331	CL:Ct (5:4)	37.5	5.93
*FnipNPR5*	FNI_icon04380981	FNI_icon04380981.1:1-919	919	919	190	Ct	21.3	5.44
*FvirNPR1*	evm.model.ctg104.806	Fvir1_FvScbg_v1.0:4416036-4423359	7324	1830	579	Nuc:Ct_Nuc (6.5:6)	64.2	6.39
*FvirNPR31*	evm.model.ctg53.483	Fvir3_FvScbg_v1.0:7054130-7058288	4159	2628	587	CL:Ct:ER(3:3:3)	65.3	6.46
*FvirNPR32a*	evm.model.ctg16.266	Fvir6_FvScbg_v1.0:28680195-28684954	4760	2667	592	Ct	66.3	5.84
*FvirNPR32b*	evm.model.ctg16.265	Fvir6_FvScbg_v1.0:28677873-28679626	1754	978	172	Nuc:Nuc_Pm (7: 5.5)	19.3	5.47
*FvirNPR33*	evm.model.ctg16.264	Fvir6_FvScbg_v1.0:28671131-28675723	4593	2056	578	Nuc	64.7	5.71
*FvirNPR5*	evm.model.ctg10.1018	Fvir6_FvScbg_v1.0:17711496-17714838	3343	2466	506	CL:ER(4:3)	55.1	6.30
*FveNPR1*	FvH4_1g07810	Fvb1_v4.0.a1:1g4137661-4139975	2315	2001	595	Ct_Nuc:Nuc:Ct (5:4.5:4.5)	66.1	6.56
*FveNPR31*	FvH4_3g11950	Fvb3_v4.0.a1:3g7057867-7062282	4763	3137	559	Nuc:Ct_Nuc (6.5:5.5)	62.5	6.62
*FveNPR32a*	FvH4_6g38830	Fvb6_v4.0.a1:6g30715984-30720865	5163	3125	591	Nuc	65.9	6.21
*FveNPR32b*	FvH4_6g38821	Fvb6_v4.0.a1:6g30713812-30715119	1308	510	169	Ct	18.9	6.52
*FveNPR33*	FvH4_6g38820	Fvb6_v4.0.a1:6g30706741-30711990	5250	2720	578	Nuc	64.6	5.77
*FveNPR5*	FvH4_6g25090	Fvb6_v4.0.a1:6g19041447-19045465	4019	3135	506	CL:ER(4:3)	55.1	6.30
*FxaNPR1a*	FxaC_1g08070	Fvb1-4:1g3379833-3381904	2072	1758	585	Nuc:Ct_Nuc (6.5:6.5)	65.0	6.31
*FxaNPR1b*	FxaC_3g07950	Fvb1-3:3g3702237-3704301	2065	1758	585	Nuc:Ct_Nuc:Ct (6.5:6.5:5.5)	64.8	6.34
*FxaNPR1c*	FxaC_2g04070	Fvb1-2:2g1987750-1989813	2064	1758	585	Nuc:Ct_Nuc:Ct (6.5:6.5:5.5)	64.8	6.31
*FxaNPR1d*	FxaC_2g12680	Fvb1-2:2g5651753-5653817	2065	1758	585	Nuc:Ct_Nuc:Ct (6.5:6.5:5.5)	64.9	6.74
*FxaNPR1e*	FxaC_4g32730	Fvb1-1:4g24082443-24086784	4342	1443	480	CL:Ct_Nuc(6:4.5)	53.1	6.17
*FxaNPR31a*	FxaC_12g38540	Fvb3-1:12g25671777-25676553	4777	3253	587	CL:Ct:Vc(3:3:3)	65.4	6.69
*FxaNPR31b*	FxaC_11g10120	Fvb3-3:11g4779868-4784455	4588	3087	587	Ct:Nuc(4:3)	65.3	6.39
*FxaNPR31c*	FxaC_10g12880	Fvb3-2:10g6319585-6324148	4564	3026	587	CL:Ct:ER(3:3:3)	65.2	6.33
*FxaNPR32a*	FxaC_22g16240	Fvb6-3:22g7985231-7990237	5007	2939	592	Nuc	66.1	6.10
*FxaNPR32b*	FxaC_21g19040	Fvb6-1:21g8680678-8685655	4978	2943	591	Nuc	65.9	6.24
*FxaNPR32c*	FxaC_24g47651	Fvb6-4:24g28858157-28862733	4577	2505	592	Ct	66.1	6.07
*FxaNPR32d*	FxaC_23g61170	Fvb6-2:23g35711876-35716520	4645	2552	592	Ct	66.3	6.94
*FxaNPR32e*	FxaC_22g16241	Fvb6-3:22g7990555-7992532	1978	1213	163	Nuc:Nuc_Pm(6:5)	18.0	6.31
*FxaNPR32f*	FxaC_24g47650	Fvb6-4:24g28855998-28856575	578	318	105	CL	12.1	5.29
*FxaNPR32g*	FxaC_21g19041	Fvb6-1:21g8686054-8687896	1843	1051	169	Ct	18.8	6.21
*FxaNPR33a*	FxaC_21g19050	Fvb6-1:21g8689400-8694727	5328	2804	578	Nuc	64.6	5.83
*FxaNPR33b*	FxaC_23g61160	Fvb6-2:23g35703222-35708378	5157	2657	578	Nuc	64.7	5.71
*FxaNPR33c*	FxaC_22g16250	Fvb6-3:22g7993972-7999331	5360	2868	578	Nuc	64.8	5.77
*FxaNPR33d*	FxaC_24g47640	Fvb6-4:24g28849358-28854622	5265	2733	578	Nuc	64.9	5.55
*FxaNPR5a*	FxaC_21g36330	Fvb6-1:21g18274937-18277824	2888	2040	506	CL:ER(4:3)	55.1	6.28
*FxaNPR5b*	FxaC_22g32930	Fvb6-3:22g19234481-19237536	3056	2197	506	CL:ER(4:3)	55.3	6.27
*FxaNPR5c*	FxaC_23g15430	Fvb6-2:23g10470309-10473618	3310	2467	506	CL:ER(4:3)	55.2	6.20
*FxaNPR5d*	FxaC_24g32090	Fvb6-4:24g18449601-18452673	3073	2192	502	CL:ER(4:3)	54.7	6.23

^a^ The subcellular localization was predicted at the WoLF PSORT website. Nuc for nucleus; Ct for cytoplasm; Ct_Nuc for shuttle between cytoplasm and nucleus; ER for endoplasmic reticulum; CL for chloroplast; Vc for vacuolar membrane; and Nuc_Pm for shuttle between nucleus and plasma membrane. ^b^ The molecular weight of the deduced NPR proteins, values in kilo Dalton. ^c^ The isoelectric point of strawberry NPR proteins.

## Data Availability

The datasets generated during and/or analyzed during the current study are available from the corresponding author Ke Duan (kduan936@126.com) on reasonable request.

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
