# Peer review of "Unraveling NPR-like Family Genes in Fragaria spp. Facilitated to Identify Putative NPR1 and NPR3/4 Orthologues Participating in Strawberry-Colletotrichum fructicola Interaction"

_plants, 2022, doi:10.3390/plants11121589_

Round 1

Reviewer 1 Report

The manuscript identified and characterized homologues of AtNPR1-like genes from five strawberry genomes. The authors performed a detailed analysis of the phylogenetics of the genes by both nucleic acid and amino acid sequences. Experimental evidence by qPCR proved the role of FveNPR1 in resistance against the necrotroph Colletotrichum fructicola. Overall, the manuscript is well written and suitable for publication in its current form. 

Author Response

Thank reviewer for your comments. We will perform careful check to improve English language and style.

Thank you again for your valuable time and inspiring comments.

Reviewer 2 Report

The manuscript entitled “ Unraveling NPR-like family genes in Fragaria spp. Facilitated to identify putative NPR1 and NPR3/4 orthologues participating in strawberry - Colletotrichum fructicola interaction“ by Bai et al aimed to explore the importance of NPR-like proteins against various pathogens in strawberry. 

The authors identified conserved NPR family type proteins in F. vesca, F. viridis, F.iinumae, and F.ananassa, but observed substantial divergence in protein module and functional residues/motifs/domains. Next, they investigated the transcription responses of FveNPR1 and FxaNPR1a to C. fructicola, and found that strawberry regulates NPRs-centered defense systems against C. fructicola. These findings may provide valuable resources for resistance breeding in strawberries. 

Overall, the method used in the study is thorough. Conclusions are appropriate, and supported by the data. Statistical analysis is provided within the manuscript. The whole study is sound, and I recommend accepting it.

Author Response

Thank reviewer for your comments. We will carefully check our manuscript to improve it.

Thank you again for your valuable time and inspiring comments.

Reviewer 3 Report

In this manuscript, the authors investigated the presence of the salicylic acid receptors NPR-like genes in the genomes of both diploid and octoploid strawberry. Further transcriptional analysis suggested the potential roles of FveNPR1 and FxaNPR1a in response to Colletotrichum fructicola. Here, the authors identified the candidate NPR genes involving in strawberry immune resistance against C. fructicola. My comments are listed in detail as follows.

1. Figure 1, it is hard to see the bootstrap value which are too small on the figure. Some are even covered by colored dots.

2. Line 312, it should be FveNPR31-33, instead of  FvNPR31-33.

3. Figure 6B, qPCR is more convincing than RT-PCR.

Author Response

Q1. Figure 1, it is hard to see the bootstrap value which are too small on the figure. Some are even covered by colored dots.

R1: Special thanks to Reviewer for the critical and important comment. It’s really true as reviewer suggested that Figure 1 should be improved. We have corrected this illustration. Please see Page 6 Figure 1 in the corrected manuscript.

 Q2. Line 312, it should be FveNPR31-33, instead of  FvNPR31-33.

R2: Many thanks to Reviewer. Corrected. Please see Page 12 Line 353.

Q3. Figure 6B, qPCR is more convincing than RT-PCR.

R3: Thank Reviewer for the important comment. It’s true as reviewer suggested that qPCR provides a more convincing analysis of the relative gene expression. However, this method could provide a relatively quantified expression analysis of certain gene, but not a comparative expression analysis between genes, especially those members of the same family. Furthermore, it’s hard to achieve a satisfactory qPCR analysis for genes expressed at extremely low levels. By contrast, RNAseq and semi-quantitative RT-PCR could provide more information about the absolute expression for certain gene and a comparison between genes.

In current work, because FveNPR1/5 and FxaNPR1/5s were expressed at very low levels in strawberry, qPCR analysis did not produce any repeatable results. Thus RT-PCR were performed for this family and we differentiated the five homoalleles of FveNPR1 in octoploid strawberry (Figure 6B). Figure 7 displayed the qPCR analysis of FveNPR3s and FxaNPR3s in response to C. fructicola infection. Thank you for your valuable time and constructive comments.